# Zero-shot Synthetic Video Realism Enhancement via Structure-aware Denoising

## Abstract

We propose an approach to enhancing synthetic video realism, which can re-render synthetic videos from a simulator in photorealistic fashion. Our realism enhancement approach is a zero-shot framework that focuses on preserving the multi-level structures from synthetic videos into the enhanced one in both spatial and temporal domains, built upon a diffusion video foundational model without further fine-tuning. Specifically, we incorporate an effective modification to have the generation/denoising process conditioned on estimated structure-aware information from the synthetic video, such as depth maps, semantic maps, and edge maps, by an auxiliary model, rather than extracting the information from a simulator. This guidance ensures that the enhanced videos are consistent with the original synthetic video at both the structural and semantic levels. Our approach is a simple yet general and powerful approach to enhancing synthetic video realism: we show that our approach outperforms existing baselines in structural consistency with the original video while maintaining state-of-the-art photorealism quality in our experiments.

## 1 Introduction

Training models for autonomous vehicles requires vast amounts of data, particularly for rare long-tail scenarios. However, the collection and labeling of real-world video data are costly and often fail to capture these critical events. To address this challenge, researchers have turned to synthetic data generated from simulators of the environment (Agarwal et al., 2025), which offers a scalable and controllable means of producing the data required to accelerate research and development in autonomous vehicles (Hu et al., 2023; Zhou et al., 2025). A key challenge here is the domain gap between synthetic and real-world data that can introduce biases that adversely affect model performance. To alleviate this issue, we focus on enhancing the realism of synthetic videos in this work.

Initial efforts to tackle this issue used video-to-video translation techniques (Wang et al., 2018; Zhuo et al., 2022), which converted simulated driving sequences (e.g., GTA V (Richter et al., 2016)) into realistic counterparts using GAN-based methods (Goodfellow et al., 2014). These prior works demonstrated the feasibility of rendering more realistic driving footage, but they often suffered from limitations such as low resolution, poor temporal consistency, and semantic inconsistency with the original synthetic video.

Recent advancements in diffusion models (Rombach et al., 2022; Peebles & Xie, 2023) and the availability of large-scale datasets (Caesar et al., 2020; Yang et al., 2024a) have significantly enhanced generative fidelity, yielding sharper, high-resolution, and more realistic outputs. Currently, most state-of-the-art pipelines leverage controllable generation strategies, such as ControlNet (Zhang et al., 2023), where detailed simulator-derived features—including Canny edges, depth maps, semantic segmentation (Zhou et al., 2024), bird's-eye view (BEV) layouts (Wen et al., 2024; Swerdlow et al., 2024; Jiang et al., 2024), 3D boxes (Ren et al., 2025), and LiDAR (Ren et al., 2025) projections—are utilized as conditional inputs to guide the synthesis of realistic images or videos. These approaches represent the cutting edge of controllable synthetic-to-real data generation.

Despite their success, control-based methods often overlook the visual content of the source synthetic video, including aspects such as color and lighting. By regenerating appearance solely from a text prompt, they often struggle to accurately reproduce safety-critical semantic details, like the

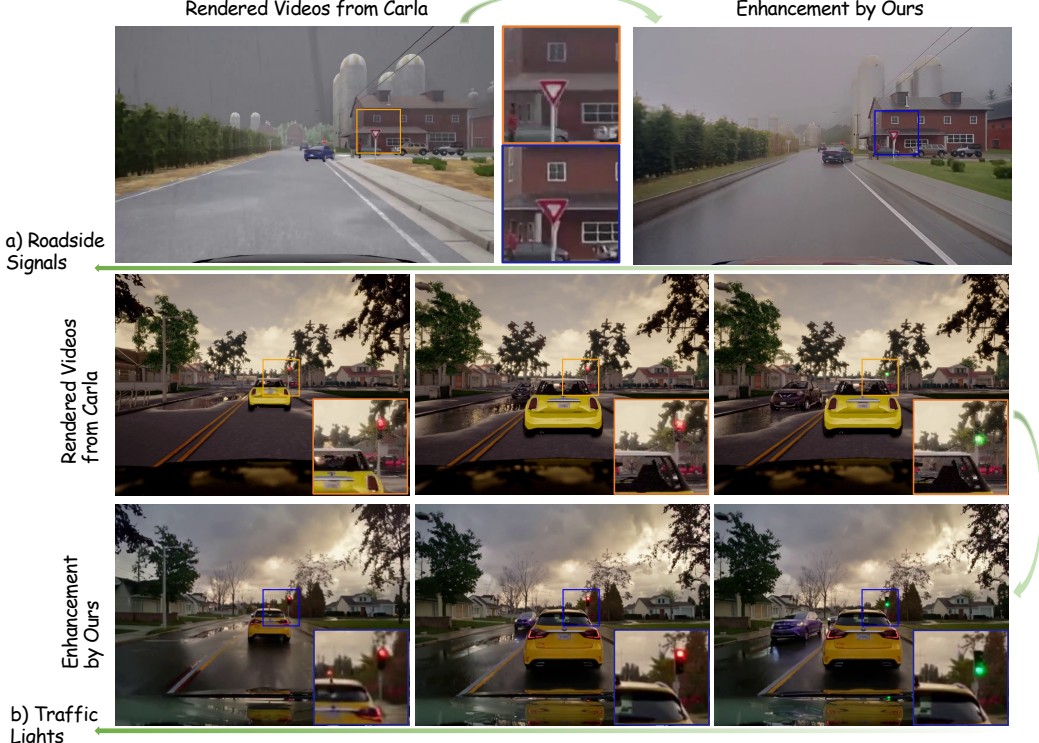

Figure 1: Enhanced videos by our structure-aware denoising method from rendered videos. Our approach could generate the videos with structural consistency and state-of-the-art photorealism quality, especially for the small objects in autonomous driving scenarios, such as roadside signals and traffic lights.

precise hue of traffic lights or the appearance of road signs. Additionally, previous GAN-based methods are constrained by limited stylistic diversity and controllability; the model can only generate generic styles it has encountered during training, rather than authentically enhancing the source video towards a desired, diverse aesthetic that aligns with real-world conditions.

Therefore, the primary challenge is to develop a method that enhances photorealism with realistic styles while preserving the essential semantic and structural content of the original video. To achieve this, we propose a framework that leverages the source video's structural information as a rich foundation for controllable enhancement. Inspired by recent advancements in video editing and generation, we introduce a new inversion-and-generation paradigm through a simple zero-shot adaptation of a state-of-the-art controllable video generation model.

Specifically, drawing from the technique DDIM Inversion (Song et al., 2020), we first invert the source video from the simulator by progressively adding noise in a deterministic manner, mapping it to an initial latent representation. This inversion process computes a specific initial noise latent that is structurally tied to the original video's content and motion. By initiating the denoising process from this content-aware latent representation—rather than from random noise—we can anchor the generation to the source semantics.

Subsequently, we apply Classifier-Free Guidance (CFG) to selectively modify the visual style during the structure-aware denoising stage. This approach enables us to eliminate the unrealistic, computer-generated textures commonly found in simulator outputs while steering the overall style towards the real-world aesthetic learned by the model. Our method significantly enhances photorealism and temporal coherence without compromising the original semantic information.

The main contributions of this work are summarized as follows:

- To the best of our knowledge, we introduce the first zero-shot structure-aware denoising framework that adapts a pre-trained video diffusion model for the photorealistic enhancement of synthetic videos, eliminating the need for domain-specific post-training.

- We propose a simple yet effective zero-shot inversion-and-generation framework that ensures semantic preservation while enhancing realism through the generation process.

- We have developed an evaluation protocol to quantify the semantic and temporal consistency of small and safety-critical objects (e.g., traffic lights, road signs), ensuring a rigorous assessment of semantic fidelity.

## 2 RELATED WORK

### 2.1 VISUAL SYNTHESIS FOR AUTONOMOUS DRIVING

In autonomous driving, large-scale data collection is expensive and time-consuming. Obtaining visual contents via simulators (Richter et al., 2016; Dosovitskiy et al., 2017a) is an easier way to build a physical world based on the various annotations. However, the domain gap between the simulator-rendered videos and the real-world contents is challenging. Richter et al. (2021), Pasios & Nikolaidis (2025a), and Pasios & Nikolaidis (2025b) trained an image enhancement network to enhance the photorealism of rendered images under the supervision of the adversarial objective. Wang et al. (2018), with its follow-up works Zhuo et al. (2022); Mallya et al. (2020) extended the generative adversarial learning framework to the video-to-video synthesis, translating the semantic maps of the synthetic videos to the photorealistic outputs with temporally coherent. With the powerful generative ability of diffusion models, Zhao et al. (2024) and Pronovost et al. (2023) utilized the diffusion-based model conditioning on the synthetic images to fill the domain gaps and improve structural fidelity. However, when the controllable diffusion-based framework extends to the video scenario, the temporal consistency remains challenging. One direction Gao et al. (2024); Yang et al. (2024a); Zhou et al. (2024) employs a two-stage pipeline: first generate a photorealistic initial image, and then extrapolate subsequent frames with a video prediction model. However, this decoupling of content generation from temporal dynamics introduces significant limitations. These models often lack fine-grained control over vehicle kinematics, such as speed, and struggle to plausibly synthesize content for disoccluded regions. Furthermore, compounding prediction errors typically restricts the output to short, temporally limited sequences. Another direction is to train a world model involving physical knowledge. Wang et al. (2024) is the first world models approach for autonomous driving and generated the realistic videos on multiple conditions, such as HD maps, 3D box, and actions. Agarwal et al. (2025) proposes Cosmos, a world foundation model platform, and Alhaija et al. (2025) leads cosmos-transfer, a customized downstream application, focused on the synthetic-to-real video transfer in robotics and autonomous driving conditioning on multi-modality inputs. In this paper, we are based on the cosmos-transfer and aim to enhance the video realism through the generation process while ensuring semantic preservation.

### 2.2 CONTROLLABLE TEXT-TO-VIDEO DIFFUSION MODELS

Text-to-video generation aims to convert low-dimensional data, text prompts, to the high-dimensional modality, videos. The basic structure of generative models is U-Net-based models (Chen et al. (2023); Ho et al. (2022); Bar-Tal et al. (2024)) and transformer-based models (Brooks et al. (2024); Yang et al. (2024c); Wan et al. (2025); Kong et al. (2024)). With the powerful generative ability of text-to-visual diffusion models, controlling the generative results under the conditions has raised much attention in the community. Some pioneering works, such as DDIM Inversion (Song et al. (2020)), ControlNet (Zhang et al. (2023)), and IP Adapter (Ye et al. (2023)), have led the direction of controllable diffusion models, with their variants controlling different text-to-visual foundation models. DDIM Inversion is a zero-shot method that adds the noise to the condition in the reverse process and denoises the condition in the denoising process. Due to the zero-shot property, it is easy to manipulate the latents during inversion and thus lead to different controlling results. FateZero (Qi et al. (2023)) achieves remarkable zero-shot video editing results by adjusting the DDIM inversion and capturing attention maps to maintain structural consistency. ControlNet is another fine-tuning control method that generates high-fidelity results by conditioning different inputs, such as Canny, depth, or segmentation maps. Cosmos-transfer (Alhaija et al., 2025) is a controllable version based on the world foundation model, Cosmos-predict (Agarwal et al., 2025), with the DiT structure using the ControlNet technique. VACE (Jiang et al. (2025)) is the controllable version of WAN (Wan et al. (2025)) via a context adapter structure. In this paper, we combine inversion techniques with ControlNet to enhance the realism of synthetic videos through structure-aware denoising.

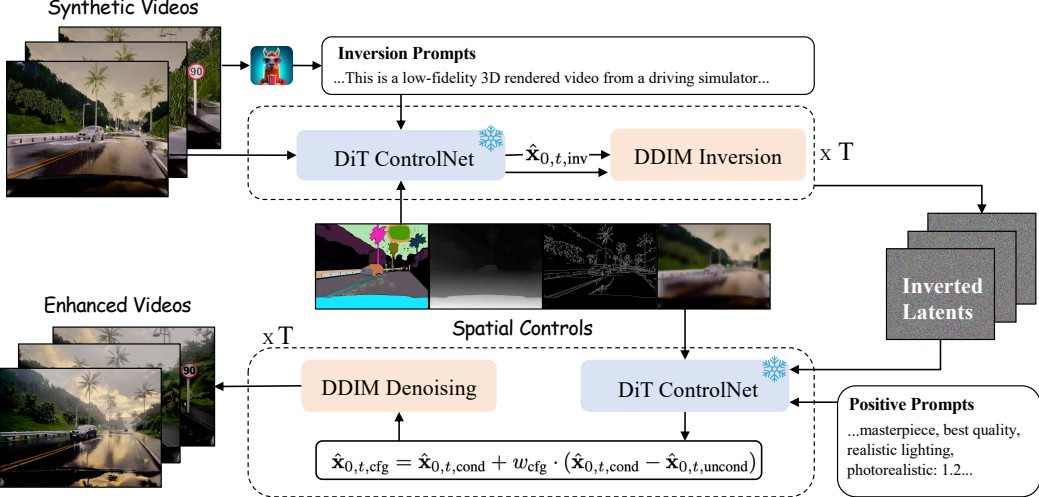

Figure 2: Overview of our pipeline on synthetic video realism enhancement. We inject DDIM inversion into the ControlNet version of the world model (Agarwal et al., 2025) in a zero-shot structure-aware denoising manner, aiming to improve structural consistency while maintaining photorealism.

## 3 METHOD

The foundation of our proposed framework is Cosmos-Transfer1 (Alhaija et al., 2025), a state-of-the-art model that addresses the synthetic-to-real domain transfer problem in video generation. Its architecture is distinguished by the integration of a Video ControlNet, which enables robust temporal and spatial conditioning.

### 3.1 PRELIMINARY: COSMOS-TRANSFER

**Multi-modal Conditioning** A key feature of Cosmos-Transfer1 is its ability to simultaneously process multiple spatial conditions. The model accepts a text prompt for semantic guidance, along with a set of three distinct spatial control maps: a depth map (Yang et al., 2024b), a segmentation map (Ravi et al., 2024), and a Canny edge map. The text prompt is converted into an embedding $\mathbf{c}_{\text{text}}$ by a text encoder, while the spatial maps are collectively denoted as $\mathbf{c}_{\text{spatial}}$.

**Structure-aware Denoising Process** The core of our generative process is a DiT-based denoising network (Peebles & Xie, 2023), $\boldsymbol{\epsilon}_\theta$, which operates in a continuous time framework. The network is trained to predict the noise component $\boldsymbol{\epsilon}$ from a noisy image $\mathbf{x}_t$. Given $\mathbf{x}_t$ at noise level $\sigma_t$ and the conditions $\mathbf{c}_{\text{spatial}}$ and $\mathbf{c}_{\text{text}}$, the predicted noise at timestep t, which we denote as $\mathbf{n}_t$, is given by:

$$\mathbf{n}_t = \boldsymbol{\epsilon}_\theta\big(\mathbf{x}_t, \sigma_t, \mathbf{c}_{\text{spatial}}, \mathbf{c}_{\text{text}}\big). \tag{1}$$

To integrate the spatial conditions $\mathbf{c}_{\text{spatial}}$, we employ a ControlNet structure that runs in parallel to the main DiT backbone. The control signals are injected into the transformer blocks, with their influence governed by an indicator function. For each block $i$, the final output feature map, $\mathbf{h}_i^{\text{final}}$, is computed as:

$$\mathbf{h}_i^{\text{final}} = \mathbf{h}_i^{\text{main}} + \mathbb{I}(i \in \{1, 2, 3\}) \cdot w_c \cdot \mathbf{h}_i^{\text{control}}, \tag{2}$$

where $\mathbb{I}(\cdot)$ is the indicator function that is equal to 1 if the condition is true and 0 otherwise. The term $w_c$ is the **control weight**. Note that we balance the influence of the spatial conditions only for the first three blocks.

**Image Sampling with Classifier-Free Guidance** During inference, Classifier-Free Guidance (CFG) (Ho & Salimans, 2022) is utilized to enhance adherence to the text prompt while steering the generation away from the negative prompt. At each timestep $t$, two noise predictions: a conditional one, $\mathbf{n}_{t,\text{cond}}$, using the target text prompt, and an unconditional one, $\mathbf{n}_{t,\text{uncond}}$, using a negative prompt are generated. Following the EDM (Karras et al., 2022) formulation, these noise predictions

are converted into corresponding *estimated clean images*, $\hat{\mathbf{x}}_{0,t,m \in \{\text{cond, uncond}\}}$, using:

$$\hat{\mathbf{x}}_{0,t,m} = \left( \frac{\sigma_{\text{data}}^2}{\sigma_t^2 + \sigma_{\text{data}}^2} \right) \mathbf{x}_{t,m} + \left( \frac{\sigma_t \sigma_{\text{data}}}{\sqrt{\sigma_t^2 + \sigma_{\text{data}}^2}} \right) \mathbf{n}_{t,m}, \tag{3}$$

where $\sigma_t$ is the noise level in the timestep $t$, and $\sigma_{\text{data}}$ is the standard deviation of the training data. These two estimates are then combined into a single guided estimate, $\hat{\mathbf{x}}_{0,t,\text{cfg}}$, by extrapolating from the unconditional towards the conditional prediction:

$$\hat{\mathbf{x}}_{0,t,\text{cfg}} = \hat{\mathbf{x}}_{0,t,\text{cond}} + w_{\text{cfg}} \cdot (\hat{\mathbf{x}}_{0,t,\text{cond}} - \hat{\mathbf{x}}_{0,t,\text{uncond}}), \tag{4}$$

where $w_{\text{cfg}}$ denotes CFG. $\mathbf{x}_{t-1}$ is then computed within an Euler sampler, as:

$$\mathbf{x}_{t-1} = \mathbf{x}_t + \left( \frac{\mathbf{x}_t - \hat{\mathbf{x}}_{0,t,\text{cfg}}}{\sigma_t} \right) (\sigma_{t-1} - \sigma_t). \tag{5}$$

## 3.2 Synthetic Video Realism Enhancement

To bridge the synthetic-to-real gap, we introduce a novel pipeline that enhances photorealism while preserving the low-frequency structural information of a source simulation. While traditional methods might directly use high-frequency conditioning maps (e.g., Canny edges, segmentation) from a synthetic image/video to guide a denoising process from pure noise, we found this often fails to retain the global composition. Our method, in contrast, employs a latent inversion and subsequent re-generation process to achieve a more faithful transfer.

**Source Simulation and Conditioning:** Given a synthetic video, we first generate the video caption using a model such as Videollama3 (Zhang et al., 2025). This model concurrently provides two text prompts: an *inversion prompt* ($\mathbf{c}_{\text{text}}^{\text{inv}}$), which accurately describes the synthetic video content, and a *positive prompt* ($\mathbf{c}_{\text{text}}^{\text{real}}$), designed to elicit photorealism. We extract the corresponding spatial conditioning maps $\mathbf{c}_{\text{spatial}}$ (depth, segmentation, Canny).

**Deterministic Latent Inversion:** The core of our contribution is to find a unique noise latent $\mathbf{x}_T$ that deterministically reproduces the synthetic video $\mathbf{x}_0^{\text{sim}}$. This is achieved by reversing the generative process through a DDIM-style inversion. Starting with $\mathbf{x}_0 = \mathbf{x}_0^{\text{sim}}$, we iteratively add noise by reversing the Euler sampler step for $t = 0, \ldots, T-1$, where $\sigma_{t+1}$ is the noise level of the next inversion step:

$$\mathbf{x}_{t+1} = \mathbf{x}_t + \left( \frac{\mathbf{x}_t - \hat{\mathbf{x}}_{0,t,\text{inv}}}{\sigma_t} \right) (\sigma_{t+1} - \sigma_t). \tag{6}$$

At each inversion step, the guided clean video latent estimate $\hat{\mathbf{x}}_{0,t,\text{inv}}$ is computed using the inversion prompt $\mathbf{c}_{\text{text}}^{\text{inv}}$ and the spatial maps $\mathbf{c}_{\text{spatial}}$. This computation follows the full generative logic: the ControlNet-integrated denoising network first predicts the noise (Eq. 1, 2), which is then used to derive the clean estimate via the EDM formulation (Eq. 3). This process effectively encodes the full structural and low-frequency information of $\mathbf{x}_0^{\text{sim}}$ into the final latent $\mathbf{x}_T$.

**Structure-aware denoising for Photorealism:** With the information-rich latent $\mathbf{x}_T$ secured, we perform the final generative step. We initiate the standard denoising process, as detailed in Section 3.1, starting from the inverted latent $\mathbf{x}_T$. However, for this forward process, we use the *positive prompt* $\mathbf{c}_{\text{text}}^{\text{real}}$ generated in the initial step. The spatial conditions $\mathbf{c}_{\text{spatial}}$ remain unchanged to ensure strict structural consistency with the original simulation.

This two-stage approach allows Cosmos-transfer1 to leverage the inverted latent to preserve content and structure while using the new positive prompt to steer the synthesis into a realistic domain, effectively bridging the synthetic-to-real gap.

## 4 Experiments

### 4.1 Datasets

To evaluate our framework, we built a custom benchmark using the CARLA simulator (Dosovitskiy et al., 2017b; Sun et al., 2022). The benchmark consists of 900 unique video sequences, each

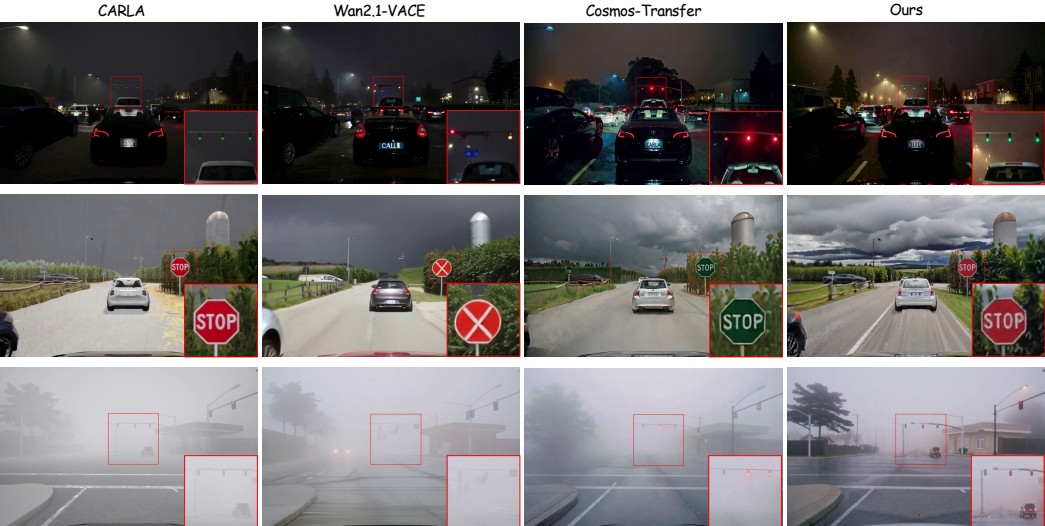

Figure 3: Qualitative results of our methods on synthetic video realism enhancement. We could maintain the photorealism and improve the structural consistency in the diverse outdoor conditions.

121 frames long, designed to rigorously test the synthetic-to-real generalization of our method. The sequences were generated under a diverse matrix of environmental conditions, systematically combining different times of day (daytime and nighttime) with various weather scenarios (clear/sunny, rainy, and dense fog). This comprehensive dataset serves as our primary testbed for performance assessment.

## 4.2 EVALUATION METHODS

**Photorealism Assessment**  To measure photorealism, we use GPT-4o (Achiam et al., 2023) to perform pairwise comparisons between our model and leading baselines, Cosmos-Transfer1 (Alhaija et al., 2025) and WAN2.1 VACE (Jiang et al., 2025). From each video pair, we randomly sample five corresponding frames. We then instruct GPT-4o to act as a graphics expert and choose the more photorealistic frame based on lighting, shadows, textures, and overall realism. We report the final score as our model's win rate, representing the percentage of times it was judged to be more realistic.

**Object and Semantic Consistency**  To quantitatively evaluate the temporal consistency and identity preservation of critical objects, we propose an object-centric feature consistency metric. For each frame $t$ in the original video ($I_t^{\text{orig}}$) and the generated video ($I_t^{\text{gen}}$), we first identify critical objects such as 'traffic lights' and 'traffic signs' using a GroundingDINO+SAM2 pipeline (Liu et al., 2024; Ravi et al., 2024), which yields a set of object masks $\{M_t^k\}$.

We then extract dense feature maps, $F_t^{\text{orig}}$ and $F_t^{\text{gen}}$, using a pre-trained DINOv2 (Oquab et al., 2023) or CLIP (Radford et al., 2021) encoder $\Phi$. The core of our metric is to measure the feature-space similarity within each object mask. Specifically, for each frame, we first compute a pixel-wise similarity map between the original and generated feature maps. Then, for each of the $K_t$ detected objects in frame $t$, we calculate its average similarity by aggregating the scores within its mask region $M_t^k$ and normalizing by the mask's area $|M_t^k|$. The final consistency score, $\mathcal{S}_{\text{consistency}}$, is the mean of these average similarities across all objects and all frames, as formulated in Equation 7:

$$\mathcal{S}_{\text{consistency}} = \frac{1}{T} \sum_{t=1}^{T} \sum_{k=1}^{K_t} \frac{1}{|M_t^k|} \sum_{p \in M_t^k} s\left(F_t^{\text{orig}}(p), F_t^{\text{gen}}(p)\right) \tag{7}$$

where $s(\cdot, \cdot)$ is the chosen similarity function (e.g., cosine similarity). A higher $\mathcal{S}_{\text{consistency}}$ score indicates better preservation of object appearance and identity.

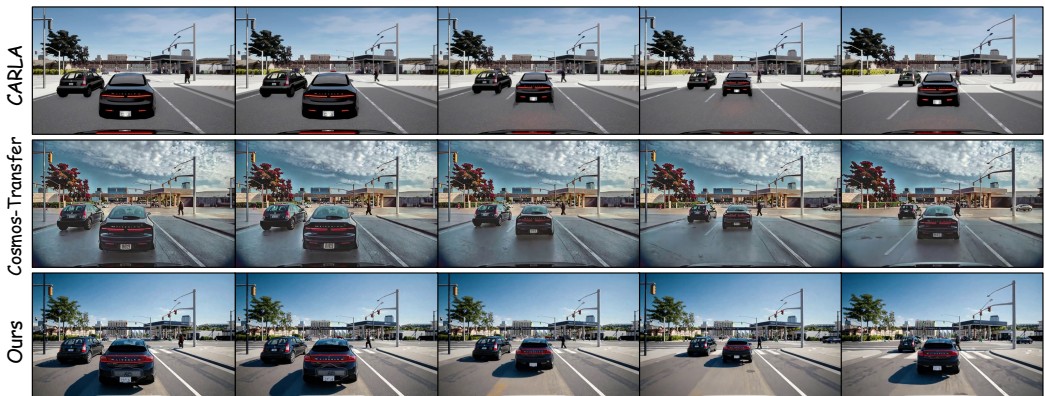

Figure 4: Examples of our method on temporal alignments. Our method could maintain the change of traffic lights simultaneously as well as improve the photorealism of the enhanced videos, such as the shadows of cars. *Best view to zoom in.*

**Perceptual Similarity (LPIPS)** We utilize the Learned Perceptual Image Patch Similarity (LPIPS) metric (Zhang et al., 2018) to measure the perceptual distance between the generated frames and the source frames. We report the average LPIPS score across all frames in a video.

**Video Quality** We use VBench (Huang et al., 2024) to evaluate the temporal consistency and overall quality of our generated videos. We observe noticeable structural artifacts: objects often deform or change shape over time using frame-by-frame generation. This lack of structural integrity is penalized by VBench, leading to lower scores on metrics such as "Dynamic Degree" and "Imaging Quality." In contrast, video-generation–based methods (e.g., WAN2.1-VACE and Cosmos-Transfer1), which model temporal dynamics directly, avoid these issues and produce more temporally coherent object structures.

### 4.3 QUALITATIVE RESULTS

To provide a visual and intuitive assessment of our method's efficacy, we present a qualitative comparison against two representative state-of-the-art baselines: WAN2.1 VACE (Jiang et al., 2025) and Cosmos-Transfer1 (Alhaija et al., 2025) on CARLA. Our analysis, illustrated in Figure 3, focuses specifically on the synthesis fidelity of safety-critical objects, namely traffic lights and roadside signs, which are common failure points for existing synthetic-to-real generation pipelines. We apply our methods to GTA videos (Richter et al., 2016), and Figure 5 shows the enhanced results.

**Traffic Light Fidelity** As shown in the top rows of Figure 3, the baseline methods struggle to maintain the semantic integrity of traffic lights. VACE, for instance, often produces results where the light's color is either desaturated, incorrect (e.g., generating a greenish hue for a red light), or appears as an indistinct blur. Similarly, Cosmos-Transfer1, while preserving structure, frequently fails to render the correct color state, undermining the object's critical function. In stark contrast, our method faithfully preserves the original semantic state and color from the simulator input. It is worth mentioning that our methods could outperform the baselines on small object consistency spatially and temporally, even in challenging circumstances, such as night or foggy weather. In addition, our model renders a vibrant and unambiguous red light that remains temporally consistent across frames, demonstrating its ability to retain crucial appearance information that other methods discard, which is shown in Figure 4.

**Roadside Sign Consistency** The challenge of rendering small, detailed objects is further highlighted in the comparison of roadside signs (The second row of Figure 3). These signs often contain important symbols or text that must be identifiable. The baselines tend to render these signs with garbled textures and distorted shapes, making them unrecognizable. For example, a stop sign might lose its distinct octagonal shape or its iconic white lettering. Our approach, however, maintains the sign's structural integrity, color, and key visual features. By leveraging the complete information from the source video via our inversion-regeneration pipeline, our method ensures that the sign remains a distinct and identifiable object, even at a distance.

| Methods | LPIPS↓ | Photorealism | Video Quality | | Small Object Alignment | |
| | | Votes by MLLMs ↑ | Dynamic Degree ↑ | Imaging Quality ↑ | DINO↑ | CLIP ↑ |
| --- | --- | --- | --- | --- | --- | --- |
| CARLA (Dosovitskiy et al., 2017b) | - | 0% | - | - | - | - |
| FLUX-controlnet (frame by frame) | 0.5725 | 58.6% | 0.363 | 0.504 | 0.481 | 0.697 |
| Wan2.1-VACE (Jiang et al., 2025) | 0.4137 | 46.7% | 0.69 | 0.576 | 0.513 | 0.689 |
| Cosmos-Transfer (Ren et al., 2025) | 0.4184 | 51.7% | 0.69 | 0.654 | 0.529 | 0.706 |
| Ours | 0.3683 | Reference(50%) | 0.71 | 0.644 | 0.550 | 0.751 |

Table 1: Quantitative results of our methods on CARLA. We pairwisely compare our method to the baselines, using our method as the reference with a 50% vote rate for photorealism evaluation.

## 4.4 QUANTITATIVE RESULTS

We benchmarked our method against several key baselines, with results summarized in Table 1. In a direct comparison with an alternative approach that also generates frames sequentially using Flux Multicontrolnet (Labs, 2024), our method delivers markedly superior temporal consistency and video quality. Concurrently, when evaluated against our base model, Cosmos-Transfer1 (Alhaija et al., 2025), our technique provides enhanced object consistency and perceptual similarity while preserving a comparable degree of photorealism. These results highlight our method's advantages over both similar frame-by-frame architectures and our foundational model.

**Photorealism Assessment** To assess photorealism, we conducted a pairwise comparison using Multimodal LLMs, GPT-4o (Achiam et al., 2023) as an expert evaluator. We randomly select 100 videos from our test dataset and compare each baseline with our method. Our method, as the reference with 50% vote rate, has comparable photorealism with Cosmos-Transfer1, indicating that we maintain the realism of our base model as well as improving the consistency spatially and temporally. Although text-to-image generation models have higher realism, the flickering issues still bother, since the generated frames are not coherent if the method applies the text-to-image model frame by frame, conditioning on the synthetic videos. The results, shown in Table 3, also indicate that our method is highly competitive with WAN2.1 VACE.

**Small Object Alignment and Semantic Consistency** The most significant advantage of our method is demonstrated in its handling of small object alignment and semantic consistency. As reported in Table 1, we measured feature alignment using both DINO and CLIP scores. Our method achieves state-of-the-art results, marking a substantial improvement over both Cosmos-Transfer1 and Wan2.1-

| Methods | LPIPS↓ | Photorealism | Small Object Alignment | |
| | | Votes by MLLMs ↑ | DINO↑ | CLIP ↑ |
| --- | --- | --- | --- | --- |
| CFG=3 | 0.3611 | 31.9% | 0.562 | 0.762 |
| CFG=7 | 0.3683 | Ref(50%) | 0.550 | 0.751 |
| CFG=11 | 0.4451 | 57.8% | 0.526 | 0.713 |

Table 2: Ablation Study on Classifier-free Guidance. We apply seven as the CFG value for our final method.

VACE. These superior scores quantitatively prove that our approach preserves the semantic identity and appearance of small, critical objects with significantly higher fidelity, avoiding the color distortion and texture degradation issues that plague other methods.

**Perceptual Similarity (LPIPS)** In terms of overall perceptual similarity to the source video, our method also leads in performance. We report an average LPIPS score of 0.3683, outperforming both Wan2.1-VACE (0.4137) and Cosmos-Transfer1 (0.4184). A lower LPIPS score signifies that our generated video remains perceptually closer to the original content's structure and layout. This result is crucial, as it shows our method enhances realism and consistency without catastrophically altering the underlying scene, striking an optimal balance between faithful content preservation and stylistic transformation.

**Video Quality** Compared to generating videos frame-by-frame with FLUX multi-controlnet, which is penalized for temporal artifacts like inter-frame object deformation and motion incoherence, employing a dedicated video generation model is a clearly superior choice. Such models are inherently designed to maintain temporal consistency, which effectively eliminates the flickering and structural instability that degrade the quality of frame-by-frame synthesis.

In summary, the quantitative results unequivocally demonstrate the effectiveness of our proposed framework. While maintaining a level of photorealism that is on par with the leading baselines,

our method's key superiority lies in its preservation of semantic integrity and perceptual fidelity. It excels at maintaining the consistency of small, safety-critical objects, a crucial capability where previous methods show clear limitations.

### 4.5 ABLATION STUDY

**Classifier-free Guidance** We performed an ablation study on the Classifier-Free Guidance (CFG) scale to find the optimal balance between photorealism and semantic preservation. We tested CFG scales of 3, 7, and 11 during the denoising stage, with results in Table 2.

The study reveals a clear trade-off. While a higher CFG scale boosts subjective photorealism, it simultaneously degrades content fidelity. As CFG increases, small object alignment scores consistently decline, and the LPIPS score worsens from 0.3611 to 0.4451. This shows that stronger guidance, in its pursuit of realism, deviates from the ground truth and alters the identity of critical objects. As a result, we identified CFG=7 as the optimal setting, as it strikes an ideal balance. It achieves good photorealism performance while maintaining good object alignment and high perceptual fidelity (0.3683 LPIPS). Consequently, a CFG scale of 7 is used in all our experiments.

**Multi-condition and Control Methods** We analyze the key components in our framework: the multi-condition ControlNet and the DDIM inversion process. We conducted an ablation study, with results presented in Table 3, to understand how each part affects video quality. When we add blurred video as an additional condition, it achieves the best alignment scores. However, its photorealism score collapses to a mere 19.2%. This is because the blurred video contains the original video's style, which decreases the video's photorealism. When we remove the edge map from the condition list, the LPIPS score increases, and the alignment with synthetic videos slightly decreases.

| Methods | LPIPS↓ | Photorealism ↑ | | Small Object Alignment | |
| --- | --- | --- | --- | --- | --- |
| | | Votes by MLLMs ↑ | | DINO↑ | CLIP ↑ |
| Multi-condition | | | | | |
| Ours -canny | 0.3906 | 46.5% | | 0.541 | 0.732 |
| Ours +blur | 0.3181 | 19.2% | | 0.583 | 0.776 |
| Ours | 0.3683 | Ref(50%) | | 0.550 | 0.751 |
| Control methods | | | | | |
| Ours w/o ControlNet | 0.3820 | 4.2% | | 0.546 | 0.754 |
| Ours | 0.3683 | Ref(50%) | | 0.550 | 0.751 |

Table 3: Ablation study on multi-conditions and control methods. Our method utilizes conditions, including semantic maps, Canny edges, and depth maps.

Conversely, the inversion without ControlNet experiment demonstrates the necessity of structural guidance during inversion. The removal of ControlNet guidance leads to a catastrophic loss of realism. Our full model with the three conditions–segmentation map, depth map, and edge map–achieves the best overall performance by striking an optimal balance. We demonstrate that the synergy of DDIM inversion for content preservation and multi-condition ControlNet for structural guidance is essential for video realism enhancement.

## 5 DISCUSSION

**Limitation and future work** First, it is constrained by the base model's fixed inference window (121 frames), requiring a chunk-based approach for longer videos, which can introduce temporal discontinuities at the boundaries. Secondly, as a zero-shot model, it is sensitive to text prompts that conflict with the source video, which may occasionally produce small visual artifacts. Future work will focus on validating the downstream utility of our method and determining if enhancing synthetic data from simulators with our approach can effectively bridge the synthetic-to-real gap and benefit for training autonomous systems.

**Conclusion** In this work, we present a simple yet effective zero-shot framework for enhancing the photorealism of synthetic data, built upon existing large-scale video generation models. Crucially, our approach eliminates the need for any intensive, task-specific training. By leveraging diffusion-based inversion techniques, our method successfully preserves both the structural integrity and the semantic identity of the source video, ensuring high-fidelity enhancement. Furthermore, we introduce a novel methodology for evaluating object-level consistency within synthetic data, addressing a key challenge in generative evaluation. We believe our work provides valuable insights and a practical pipeline for data generation in autonomous driving.

## 6 ETHICS STATEMENT

This work complies with the ICLR Code of Ethics. All datasets (including Sun et al. (2022)) were obtained and used in accordance with their respective licenses and guidelines, without violating privacy. We took proactive steps to mitigate bias and prevent discriminatory outcomes. No personally identifiable information was used, and no experiments posed privacy or security risks. Transparency and integrity are our top priorities in the research process.

## 7 REPRODUCIBILITY STATEMENT

We prioritize reproducibility. Upon publication, we will release the full codebase along with instructions or scripts to obtain all datasets where licenses permit. We provide environment specifications, fixed random seeds, and complete details on architectures, hyperparameters, data splits, and training schedules. Exact evaluation scripts and metrics are included. Hardware settings and run-time notes are reported to aid replication. These resources are documented with step-by-step commands to enable straightforward reproduction of our results.

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

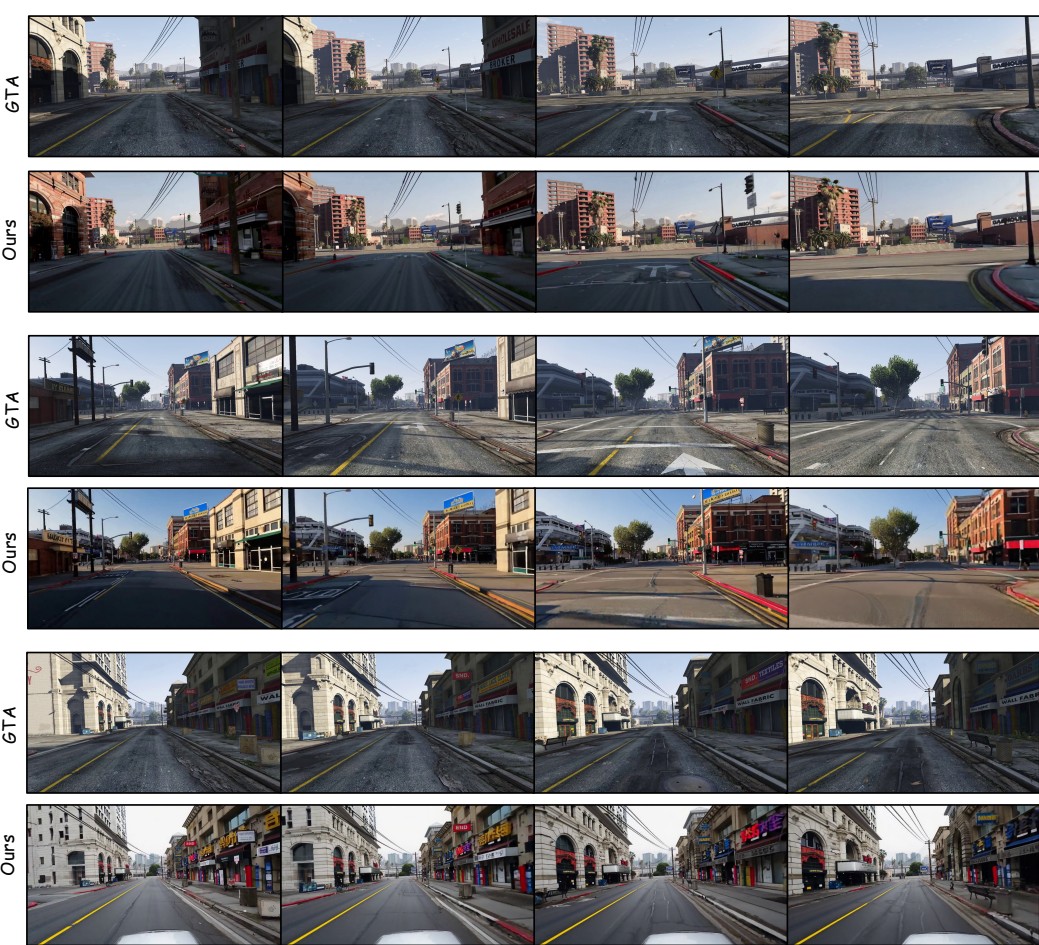

Figure 5: More examples of our methods on synthetic videos from GTA (Richter et al., 2016). We show enhanced videos every 20 frames temporally. We could maintain the photorealism and improve the structural consistency in the diverse outdoor conditions.

# A APPENDIX

## A.1 MORE QUALITATIVE RESULT

We also test our method on GTA (Richter et al., 2016) synthetic videos downloaded from the website. Examples are shown in Figure 5. We could maintain the photorealism and improve the structural consistency in the diverse outdoor conditions. The videos are shown in the supplementary materials.

