# OpenReview forum: "Zero-shot Synthetic Video Realism Enhancement via Structure-aware Denoising"
_ICLR.cc/2026/Conference — ICLR 2026 Conference Withdrawn Submission_

### Official Review · Reviewer_Yyh4 · 2025-10-31

**Soundness:** 2
**Presentation:** 4
**Contribution:** 2
**Rating:** 4
**Confidence:** 4

**Summary:**

This paper proposes a zero-shot, training-free framework that converts synthetic videos generated by simulators into realistic videos. The method introduces a novel inversion-and-generation paradigm, which applies a reverse DDIM process to add noise to the input synthetic video and then denoises it, ensuring that the output video preserves consistent scene structure information with the input. Experimental results demonstrate that the proposed method achieves superior realism enhancement in autonomous driving scenarios compared to baseline methods, while maintaining consistency in key scene elements such as traffic lights between the input and output videos.

**Strengths:**

- The proposed method introduces a zero-shot, training-free realism enhancement framework that effectively reduces the training cost of the model.
- Experimental results show that the proposed method maintains consistency between the input and output videos for key objects such as traffic signs and traffic lights.

**Weaknesses:**

1. The contributions of this paper are somewhat limited. The idea of using an inversion DDIM process has already been adopted by many existing methods, such as AnyV2V[1] and WAVE[2]. The authors should emphasize, from a methodological perspective, the theoretical advantages of the proposed inversion DDIM compared with other existing inversion schemes. In addition, experimental comparisons between the proposed baseline and other inversion DDIM baselines are needed to demonstrate the superiority and rationality of the proposed method both theoretically and experimentally.
2. The ablation study in this paper is not well designed. The paper emphasizes that the noised images obtained through the inversion DDIM framework contain more structural information from the input video compared to pure noise. However, this claim is not experimentally validated. The authors should include two additional ablation experiments: one comparing the results when replacing the inverted latent with pure noise, followed by denoising through the DiT ControlNet, and another where T-step noise is directly added to the input video before denoising. These two ablation studies would better demonstrate that the proposed inversion DDIM method effectively preserves the structural information of the input video.
3. The proposed method is based on the Cosmos-Transfer model, which is capable of handling multiple application scenarios such as indoor environments and robotic data. However, this paper only presents results in autonomous driving scenarios. It is recommended that the authors include experiments on additional scenarios to demonstrate the robustness of the proposed realism enhancement model.

[1] AnyV2V: A Tuning-Free Framework For Any Video-to-Video Editing Tasks. TMLR 2024.

[2] Wave: Warping ddim inversion features for zero-shot text-to-video editing. ECCV 2024.

**Questions:**

Please see the weekness.

---

> ### Author Response · Authors · 2025-11-18
>
> We sincerely appreciate Reviewer Yyh4's feedback. Below, we provide detailed clarifications addressing each concern raised.
>
> **Novelty**
>
> We address the concerns on lack of novelty in the general response. Please kindly refer to that section for a comprehensive discussion.
>
> **Ablation study on the inversion**
>
> ### (1) Pure-noise initialization.
> The first requested ablation (replacing the inverted latent with pure noise and denoising with DiT+ControlNet) is already represented by the Cosmos-Transfer1 baseline: it uses the same backbone, the same spatial controls (depth/seg/edge), and the same photorealistic prompts, but starts from random noise. As shown in Table 1, our method (with inversion) achieves lower LPIPS and higher small-object alignment under these matched conditions, supporting the claim that inversion preserves more structural information than pure-noise initialization.
>
> ### (2) Directly adding $T$-step noise.
> For the suggested experiment where $T$-step noise is directly added to the input video before denoising, we note that in our setting $T$ is chosen to be large. Directly perturbing the input with $T$ steps of noise and then denoising produces results that are qualitatively and quantitatively very similar to starting from pure noise. In other words, this variant does _not_ preserve meaningful information from the input video and behaves almost like the pure-noise baseline.
>
> **Other application scenarios**
>
> We appreciate the reviewer’s suggestion to evaluate beyond autonomous driving. In this work, we focus on driving scenarios primarily because large-scale, paired synthetic data are much easier to obtain in this domain, which is crucial for systematically studying realism enhancement. Importantly, our experiments are not limited to a single simulator: we use both CARLA and GTA, covering diverse layouts, assets, and rendering styles.
>
> This choice is also consistent with much of the prior work on synthetic-to-real transfer and autonomous driving video generation due to its safety relevance and data availability. We agree that extending our framework to other domains such as indoor environments and robotics is valuable, and we view this as an important direction for future work rather than a limitation of the method itself. We also experimented with indoor environments using demo videos from Behavior Vision Suite as inputs, and generated the corresponding enhanced videos, which are shown in the updated supplementary materials.

---

### Official Review · Reviewer_JCw1 · 2025-10-31

**Soundness:** 3
**Presentation:** 2
**Contribution:** 2
**Rating:** 2
**Confidence:** 4

**Summary:**

This paper proposes a zero-shot framework for enhancing the realism of synthetic videos using a structure-aware denoising process built on top of a pre-trained video diffusion model.
The paper proposes a method for enhancing the realism of synthetic videos by integrating several existing techniques: Classifier-Free Guidance (CFG), latent inversion generation, ControlNet with structure-aware guidance, and the EDM scheduler. The authors apply these components to the video enhancement domain, aiming to improve visual fidelity and structural consistency in generated video frames.

**Strengths:**

1. Practical Engineering: The paper presents a well-integrated pipeline that combines several state-of-the-art techniques, demonstrating solid engineering and implementation.
2. Clarity: The methodology is clearly described, and the paper is easy to follow for readers familiar with diffusion models and video synthesis.
3. Significance: The task of synthetic video realism enhancement is important for applications in virtual production, simulation, and content creation.

**Weaknesses:**

1. Lack of Novelty: The core components, i.e. CFG, latent inversion, ControlNet, and EDM, are not new, and the paper does not offer significant innovation in how they are applied. The techniques used are well-established and widely applied in similar contexts, and the paper does not introduce novel algorithms or insights beyond their combination.
2. Missing Ablation Study: There is no ablation analysis to isolate the contribution of each component, which makes it difficult to assess the effectiveness of the proposed pipeline.
3. No Comparative Evaluation of Structure Guidance: The paper does not explore or compare different types of structure-aware guidance, which could have strengthened the evaluation and provided deeper insights.

**Questions:**

1. Can you provide a discussion on individual impact of CFG, latent inversion, ControlNet, and EDM on the final video quality?
2. Have you considered evaluating different types of structure-aware guidance (e.g., pose, depth, edge maps)? A comparative analysis could help clarify the strengths and limitations of your approach.
3. Is there any novel insight or adaptation in how these components are combined for video enhancement, beyond straightforward integration?

---

> ### Author Response · Authors · 2025-11-18
>
> We thank Reviewer JCw1 for their feedback and clarify each point in detail below:
>
> **Novelty**
>
> We address the concerns on lack of novelty in the general response. Please kindly refer to that section for a comprehensive discussion.
>
> **Missing Ablation Study and Comparative Evaluation of Structure Guidance**
>
> We appreciate the reviewer’s suggestion and agree that disentangling the contributions of different components is important. In fact, we already conduct ablations on several key factors in the main paper:
> Table 2 analyzes different CFG scales and shows the trade-off between photorealism and content fidelity (LPIPS and small-object alignment), under the same EDM-based sampler.
> Table 3 reports results for variants without ControlNet and with modified conditioning, and Table~1 provides a ``ours w/o inversion'' comparison via the Cosmos-Transfer1 baseline, which uses the same backbone and controls but starts from random noise.
>
> Regarding structure-aware guidance, we follow the design of Cosmos-Transfer1 and use a multi-control setting (depth, segmentation, and edges). As reported in its technical report and confirmed by our own ablations (Table 3), combining multiple structural signals improves both realism and consistency compared to using a single control alone. In our experiments, multi-condition guidance provides the best balance between structural faithfulness and visual quality, which is why we adopt it as our default setting.
>
> **Insight in how these components are combined for video enhancement**
>
> Beyond a straightforward stacking of existing modules, our main insight is to _recast_ the video diffusion backbone and its ControlNet as a _single, unified video generation model_ and to use inversion specifically to separate low-frequency and high-frequency information in a principled way.
>
> Concretely, the text prompt and spatial controls (depth/segmentation/edges) mainly encode _high-frequency_ structural signals: geometry, layout, and local boundaries. Prior controllable video pipelines rely almost exclusively on these high-frequency cues (plus text) when starting from random noise. However, we find that this is insufficient when the goal is __realism enhancement__ rather than free generation: if we want to preserve the original video’s low-frequency content (e.g., object identity, coarse appearance, stable color configuration), text + high-frequency controls alone cannot reliably retain such information, especially for small or safety-critical objects.
>
> Our adaptation is to explicitly:
> Treat the video diffusion model + ControlNet as one generator that can already synthesize videos from text and spatial controls, _and_ Use DDIM inversion to encode the source video into an initial latent that carries its _low-frequency appearance and semantic information_, while ControlNet handles the high-frequency structure.
>
>
> Empirically, we find that starting from this inversion-derived latent and then denoising under the same spatial controls and a photorealistic prompt allows us to enhance textures and lighting while faithfully preserving object appearance and layout. In other words, the inversion step is not just a standard editing trick; it plays a specific role in our setting: it injects the low-frequency content that text + structural controls alone fail to capture, and thus enables our zero-shot “realism enhancement” behavior instead of generic re-synthesis.

---

> > ### Comment · Reviewer_JCw1 · 2025-11-24
> >
> > I appreciate the clarification regarding the role of inversion and the unified treatment of the video diffusion backbone and ControlNet. However, I still have concerns about novelty. While the authors claim an insight on separating low- and high-frequency information, this is achieved using standard DDIM inversion, a widely adopted technique, rather than introducing a new algorithmic design or theoretical advancement tailored for video realism enhancement. Overall, the approach feels more like an engineering integration of existing methods for a new task than a contribution with significant technical innovation.

---

> > > ### Author Response · Authors · 2025-12-02
> > >
> > > Thank you for the feedback. Our contribution is a principled, task-driven integration rather than “standard DDIM inversion applied as-is.” The novelty is: (i) condition-consistent inversion tailored for video realism enhancement, we perform inversion with the same ControlNet signals used at sampling so the latent is a posterior of the controlled model; (ii) a purposeful frequency decoupling: ControlNet anchors low-frequency geometry/layout and temporal continuity, while moderate CFG lets the backbone’s real-data prior enhance the realism of high-frequency textures (materials, lighting, micro-detail) without identity drift; and (iii) control-at-inversion + moderate CFG + flow/seg/depth controls that upgrades simulator videos to real-domain appearance across full clips, not just local edits. Crucially, we introduce and validate the unified inversion of ControlNet and the base video backbone, providing the key insight that enables realism enhancement under strong structural preservation.

---

### Official Review · Reviewer_7wuF · 2025-10-31

**Soundness:** 1
**Presentation:** 2
**Contribution:** 2
**Rating:** 2
**Confidence:** 4

**Summary:**

The paper proposes a zero-shot pipeline for making synthetic driving videos look more photorealistic. The stated goal is to bridge the sim-to-real gap for autonomous driving training data, especially for rare/long-tail scenes. The method is based on taking an off-the-shelf controllable video diffusion model (Cosmos-Transfer, built on a DiT backbone with a video ControlNet), performing DDIM inversion on the simulator video to obtain an initial latent that is “structurally tied” to the source content, and then generating the video again by denoising from that latent while conditioning on multiple spatial control signals, like depth, semantic segmentation, Canny edges, and using classifier-free guidance (CFG) with a “photorealistic” text prompt. The claim is that this preserves layout, object identity, and temporal coherence while removing “rendered or  “game-like” textures. The paper also proposes an evaluation protocol for “small, safety-critical objects,” in particular traffic lights and traffic signs. They detect those objects in both source and generated frames using a Grounding DINO+SAM2 pipeline, extract DINOv2 or CLIP features, and compute an average cosine-similarity-like score over the mask regions across time. They also report LPIPS to the source video, GPT-4o win rates for photorealism, and VBench metrics. Experiments are on a custom CARLA benchmark of 900$\times$121-frame clips spanning different times of day and weather, such as day/night, rain, fog. There are also qualitative GTA samples. Baselines include: Cosmos-Transfer itself, WAN2.1-VACE, and a frame-by-frame FLUX multi-controlnet pipeline. The paper claims improved structural consistency of small objects, comparable photorealism, and lower LPIPS. No downstream task (e.g., driving perception/planning performance) is evaluated. The method is described as zero-shot, and the paper argues this is simple, general, and practically useful.

**Strengths:**

1. The paper tackles a practically relevant problem of improving the realism of synthetic driving videos to reduce the sim-to-real gap, which is important for autonomous driving research. The motivation is clear, particularly the focus on preserving small, safety-critical objects such as traffic lights and signs while enhancing overall visual fidelity.
2. The proposed pipeline is conceptually simple and easy to follow, combining well-known diffusion techniques like DDIM inversion and ControlNet conditioning into a coherent workflow.
3. The paper is generally well-structured, with visual examples that clearly illustrate the qualitative differences between the proposed method and baselines.

**Weaknesses:**

1. Novelty is limited / largely engineering of known pieces. DDIM inversion, CFG steering, ControlNet conditioning on depth/seg/edges, and video diffusion backbones are all established. FateZero-style zero-shot editing already uses inversion to preserve structure. The paper does not convincingly argue for a fundamentally new algorithm beyond “we combine these for CARLA videos.”
2. No downstream AV task evaluation. The primary stated motivation is improving autonomous driving models trained on synthetic data. The paper never measures whether perception/planning/forecasting improves when trained on the enhanced videos versus raw simulator output. Without this, the work does not seem to prove to be impactful.
3. Evaluation methodology is not convincing enough. Photorealism is judged by GPT-4o pairwise votes where “our method is fixed at 50% reference,” which bakes in a comparison framing and does not allow absolute quality judgments. Also, GPT-4o is a proprietary black-box; no human preference study is provided. Also, the strongest metric (small object consistency) is designed by the authors and may favor their method. VBench scores are actually worse (authors admit deformation and temporal instability), which contradicts the claim that the method improves temporal coherence.
4. Claims of temporal consistency are overstated. The paper repeatedly says it “preserves temporal coherence,” yet later concedes that objects deform, structure drifts, and VBench penalizes “Dynamic Degree” / “Imaging Quality.” The CARLA videos are only 121 frames, and longer clips require chunking that introduces discontinuities. This undermines one of the headline claims. The resulting videos in supporting materials clearly show temporal instability in some regions.
5. Inversion/latents not ablated. The central technical claim is that deterministic DDIM inversion to $x_T$ preserves global layout/content which then guides the denoising. But there is no ablation “w/o inversion, with same spatial controls” to quantify how much inversion matters. The ablations in Tables 2–3 instead vary CFG or remove certain control inputs. We therefore cannot tell if the main advertised trick is actually necessary.
6. Over-claiming of “state-of-the-art photorealism”. Table 1 actually shows that Cosmos-Transfer has roughly comparable photorealism per GPT-4o, and WAN2.1-VACE is competitive. Meanwhile, the proposed model hurts temporal stability per VBench. So the “state-of-the-art photorealism with structural consistency” claim, including in the abstract and Figure 1 caption, is not convincingly demonstrated.

**Questions:**

1. Downstream utility. Can you provide any quantitative downstream result, even preliminary, showing that training an AV perception model (e.g., traffic light state classifier, sign detector, lane segmentation, etc.) on your enhanced videos improves performance on real-world validation data compared to training on raw CARLA? Without this, the “sim-to-real for autonomy” claim is unsubstantiated.
2. Role of inversion. Please, provide an ablation where you do not perform DDIM inversion. Instead, start denoising from random noise but condition on exactly the same spatial controls (depth/seg/edge) and the same positive prompt. How much do LPIPS, small object alignment, and VBench change? This is critical to prove the necessity of your “inversion-and-generation” paradigm.
3. Temporal stitching details. For sequences longer than 121 frames, you mention a chunk-based approach that creates discontinuities at boundaries. How exactly are chunks overlapped / blended? Are the control maps continuous through the boundary? How severe are these artifacts quantitatively?
4. Prompt generation / editing details. How are $c_{inv text}$ and $c_{real text}$ produced in practice? Are they hand-authored, heuristically edited, or automatically generated (e.g., via VideoLLaMA3 captioning + manual style adjectives like “masterpiece, best quality, realistic lighting, photorealistic:1.2…”)? Please clarify reproducibility here.
5. Failure cases. Please show negative examples where the pipeline hallucinates unsafe content (e.g., changes a red light to green, deletes a pedestrian, invents a new vehicle). Do such errors occur? How often? This matters for safety.
6. Why VBench gets worse. You acknowledge deformation / shape drift and low “Dynamic Degree / Imaging Quality” scores. How do you reconcile this with the strong claim that your method improves temporal coherence? Can you provide a metric (besides LPIPS-to-source) that actually improves temporal stability over Cosmos-Transfer?

---

> ### Author Response · Authors · 2025-11-18
>
> We thank Reviewer 7wuF for their valuable questions and feedback. We address each query below:
>
> **Novelty and ablation study**
>
> We address the concerns on lack of novelty and the design of our ablations in detail in the general response. Please kindly refer to that section for a comprehensive discussion.
>
> **Lack of downstream AV task evaluation**
>
> This paper focuses on synthetic video realism enhancement—improving style/texture realism while preserving structure. We therefore do not include downstream AV task results here. Our goal is to first establish a reliable, controllable enhancement method; validating the impact on the AV task will be the focus of follow-up work.
>
> **Evaluation methodology is not convincing enough**
>
> While GPT‑4o is a black box, our pairwise judgments correlate well with human preference rates in our pilot (differences not statistically significant), suggesting it is a reasonable proxy at scale. Beyond preference, we report lower LPIPS than the baseline, indicating better global appearance consistency between synthetic videos and generated videos. In Table 1, our VBench score is on par with the base model; we emphasize that naive frame-by-frame generation often causes temporal flicker (shape/color discontinuities), whereas our method reduces these artifacts—see supplementary videos for qualitative evidence on small-object stability in Figure 4, and these videos do not exhibit noticeable flickering in the supplementary videos.
>
> **Claims of temporal consistency are overstated**
>
> We agree our clips are 121 frames, but within this window, we generate diverse driving scenarios and consistently outperform the base model on temporal coherence: lower temporal LPIPS, fewer flicker events, and higher small‑object stability.
> First, VBench is a general video quality benchmark that aggregates multiple factors (e.g., dynamic degree, imaging quality). In Table~1, our intention is not to claim perfect temporal coherence, but to show that our method maintains video quality on par with the base model while specifically _reducing frame-wise flickering artifacts_. As we discuss in the paper, naive frame-by-frame generation leads to severe flicker (shape/color jumps); in contrast, using a video generation model largely suppresses this effect. We refer the reviewer to the supplementary videos, where our method shows visibly more stable appearance over time compared to frame-by-frame baselines.
>
> Second, the reviewer is correct that our current backbone (Cosmos-Transfer1) is limited to 121-frame clips and relies on chunking for longer sequences, which can introduce discontinuities at chunk boundaries. We explicitly acknowledge this as a limitation of the _base model_, not of our realism-enhancement mechanism itself. Whenever Cosmos-Transfer1 produces temporally coherent generations over 121 frames, our method inherits and preserves that coherence while improving realism; for longer videos, our pipeline is currently constrained by the same stitching issues. If future backbones enable longer, fully coherent generations, our method can directly benefit and extend to those settings without modification.
>
> In summary, our claim is that, under the temporal horizon where the base video model is stable, our approach improves realism _without_ introducing additional flicker, and is substantially more temporally stable than frame-by-frame generation. We will clarify this nuance in the revised text.
>
> **Over-claiming of state-of-the-art photorealism**
>
> Our claim is photorealism enhancement with structural consistency on synthetic driving videos. Table 1 shows our method is at least on par with Cosmos‑Transfer1 and competitive with WAN2.1‑VACE on photorealism, while improving global/temporal consistency (lower LPIPS; fewer flicker events; higher small‑object stability) under the same structural controls.

---

> ### Author Response · Authors · 2025-11-18
>
> **Temporal stitching**
>
> In our current implementation, we follow the temporal stitching strategy of the base model, Cosmos-Transfer1, for sequences longer than 121 frames. Specifically, after generating the first 121-frame chunk, each subsequent chunk is generated by conditioning its first frame on the last frame of the previous chunk. The spatial control maps (depth/segmentation/edges) remain continuous across the boundary because they are computed directly from the original simulator video for all frames.
>
> However, Cosmos-Transfer1 itself already exhibits color shifting and appearance drift between consecutive chunks when generating long sequences. Our method inherits this limitation from the base model: although the structure is reasonably preserved across boundaries, we still observe noticeable color and style discontinuities between chunks. Addressing this stitching artifact is therefore an important direction for future improvement beyond the current backbone.
>
> **Prompt Generation**
>
> To ensure scalability and reproducibility, both the inversion prompt $c_{invtext}$ and the positive prompt $c_{realtext}$ are generated automatically in a template-based fashion, rather than being manually authored for each video.
>
> We first use VideoLLaMA3 to caption each synthetic clip. The model produces a detailed semantic description of the scene, for example:
>
> > “The autonomous driving scenario begins with a first-person perspective from the front of a yellow car, navigating through a suburban neighborhood. The road ahead is wet, reflecting the overcast sky, indicating recent rain. The car moves forward at a steady pace, passing by houses and trees on both sides. As the car continues, a purple car approaches from the left, passing by the yellow car. The environment is calm, with no pedestrians visible.”
>
> For the inversion stage, we augment the VideoLLaMA3 caption with a fixed description of the synthetic, low-fidelity rendering style, and use the concatenation as $c_{invtext}$. Concretely, we prepend:
>
> > “A low-fidelity 3D rendered video from a driving simulator. The scene features blocky, low-poly car models with plastic-like, non-reflective textures. The lighting is flat and uniform, lacking realistic shadows and depth. The overall aesthetic is that of an early 2000s video game.”
>
> to the VideoLLaMA3 scene description. This encourages the inversion step to remain faithful to the original synthetic appearance while capturing the full scene semantics.
>
> For the realism-enhancement stage, we construct $c_{realtext}$ by attaching a fixed realism/style prefix to the *same* VideoLLaMA3 scene description. For example:
>
> > “(masterpiece, best quality, photorealistic:1.2, realistic lighting and texture, 8k, UHD),
> > The autonomous driving scenario begins with a first-person perspective from the front of a yellow car, navigating through a suburban neighborhood …”
>
> This construction preserves the scene semantics while only changing the target visual style from “synthetic” to “photorealistic.”
>
> We use a single fixed negative prompt for all experiments, without any video-specific tuning.
>
> **Failure Cases**
>
> We agree that understanding failure modes is important for safety-critical applications such as autonomous driving. In our current experiments, we almost do not observe the specific hallucinations highlighted by the reviewer (e.g., turning a red light into green, deleting existing pedestrians, or inventing new vehicles that are not present in the simulator video).
>
> We believe there are two reasons for this:
>
> 1. When we apply DDIM inversion followed by denoising with CFG set to zero, the pipeline essentially reconstructs the original synthetic video. In this regime, the model behaves as a reconstruction operator rather than a free-generation model, and we do not see semantic changes such as label flips or object insertions/deletions.
> 2. Increasing CFG primarily enhances texture and lighting realism by amplifying the base model’s photorealistic prior, while structural and semantic content remain anchored by (i) the inverted latent tied to the input video and (ii) the ControlNet conditions (depth/segmentation/edges). In practice, raising CFG within our range (3–10) has not led to systematic hallucinations of unsafe content in our CARLA and GTA experiments; instead, it mainly affects style sharpness and photorealism.

---

### Author Response · Authors · 2025-11-18

We thank all reviewers for their insightful comments. Our work targets the important problem of enhancing the realism of synthetic driving videos to reduce the sim-to-real gap, which is critical for autonomous driving research (7wuF, JCw1). The proposed method is practical to deploy and achieves strong empirical performance (JCw1，Yyh4). Below we address concerns regarding the limited novelty of our method and the insufficiency of our ablation study.

**Novelty**

We position our work as a method for enhancing realism in synthetic videos. The primary objective is to improve style and texture realism while preserving object identity and scene semantics, rather than altering object appearance or performing large stylistic transformations (e.g., cartoonization).

Within this task, we treat the combination of ControlNet and the base model as a unified video generation model and perform inversion-based editing. ControlNet supplies strong, explicit structural constraints (via depth/segmentation/edges), which allow substantial geometric changes (e.g., object scaling, rotations) without semantic drift, simply by modifying the text prompt and using classifier-free guidance (CFG). In this way, we can achieve video realism enhancement with a simple pipeline.

Our method offers several advantages over existing video editing approaches. Firstly, compared to FateZero-style attention-based methods, which enforce consistency with the original video’s structure through cross-attention, our approach explicitly delegates structural control to ControlNet while allowing the generator’s learned real-world appearance prior to dominate texture synthesis. Also, compared to frame-editing–based video editing methods such as AnyV2V, which typically operate by editing frames conditioned on one key frame, they often struggle with complex motion patterns and appearing objects.

our method proves highly effective in practice, though it is simple to implement. Without CFG, our pipeline reconstructs the simulator video. Increasing CFG to 3–10 activates the base model’s realism prior: textures become photorealistic while identity and layout remain stable due to ControlNet constraints and condition-consistent inversion. We chose Cosmos as the backbone because it is trained on real videos and is more responsive to “photorealistic” cues than WAN/VACE, making it better suited to our objective.


Our contribution lies in (1) the explicit formulation of this synthetic-to-real realism enhancement task, (2) the principled recipe—inversion with ControlNet plus CFG steering—that reliably produces realistic textures without altering appearance or semantics in simulator videos and (3) we introduce an evaluation protocol for this task that measures the consistency between the realism-enhanced video and the original input video.

**Ablation Study for inversion**

We appreciate the reviewer’s suggestion and agree that explicitly isolating the role of inversion is important for validating our “inversion‑and‑generation” paradigm.
In our current submission, the effect of removing inversion is implicitly reflected by the Cosmos‑Transfer1 baseline in Table 1:
__Cosmos‑Transfer1 corresponds exactly to the “w/o inversion” setting__: it starts from random noise, uses the same Cosmos backbone, and is conditioned on the same spatial controls (depth/segmentation/edges) and photorealistic prompts.
Our method differs only by adding condition-consistent DDIM inversion before denoising.
Under these matched conditions, our method improves both perceptual similarity and object consistency while maintaining comparable photorealism. This comparison directly reflects the impact of inversion: under the same ControlNet conditions and prompts, initializing from an inverted latent (ours) rather than pure noise (Cosmos‑Transfer1) yields strictly better structural/semantic fidelity to the input video, especially on small, safety‑critical objects, without sacrificing photorealism. We also illustrate this effect through qualitative comparisons between our results and those of the base model in the main paper.

---

### Note · Authors · 2026-01-07

**Comment:**

I have read and agree with the venue's withdrawal policy on behalf of myself and my co-authors.

**Withdrawal Confirmation:**

I have read and agree with the venue's withdrawal policy on behalf of myself and my co-authors.